# Assessment of the Properties of Giant Reed Particleboards Agglomerated with Gypsum Plaster and Starch

**DOI:** 10.3390/molecules27217305

**Published:** 2022-10-27

**Authors:** Maria Teresa Ferrandez-Garcia, Antonio Ferrandez-Garcia, Teresa Garcia-Ortuño, Manuel Ferrandez-Villena

**Affiliations:** Department of Engineering, Universidad Miguel Hernández, 03300 Orihuela, Spain

**Keywords:** *Arundo donax* L., valorization, composite, panel, plant residues, waste

## Abstract

This paper analyzes the properties of composite particleboards made from a mix of giant reed with gypsum plaster and starch as binders. Experimental boards were manufactured with a 10:2 weight ratio of giant reed/gypsum plaster particles and different amounts of starch. Giant reed particles used were ≤0.25 mm. The mix was pressed at a temperature of 110 °C with a pressure of 2.6 MPa for 1, 2, and 3 h. The results showed that the boards manufactured with longer times in the press and with 10 wt.% starch achieved the best physical and mechanical properties, obtaining a modulus of rupture (MOR) of 17.5 N/mm^2^, a modulus of elasticity (MOE) of 3196 N/mm^2^, and an internal bounding strength (IB) of 0.62 N/mm^2^. Thickness swelling (TS) at 24 h of the panels was reduced from 36.16% to 28.37% when 10 wt.% starch was added. These results showed that giant reed–gypsum–starch particleboards can be manufactured with physical and mechanical properties that comply with European standards for use in building construction.

## 1. Introduction

The world is facing a shortage of wood as a row material. In the last 30 years, forest area decreased from 32.5% to 30.8% of the total land, representing a net loss of 178 million hectares [1], and it is forecasted that, by 2030, there will be a deficit of 300 million m^3^. The use of wood is only expected to increase by 30% in the energy sector within this decade [2]. The scarcity of wood has a negative influence on the performance and the quality of the commercial boards. In this present situation, the construction and furniture industries must find a reliable substitute to traditional wood.

The most common wood board on the market is particleboard, which is obtained by binding wood particles with an adhesive, usually urea–formaldehyde (UF). In the construction sector, several types of particleboards include gypsum plaster as an inorganic glue (gypsum plasterboard—GPB). These panels have the advantages of no formaldehyde emission, combustion blocking off, and less loss of heat exchange [3]. However, they also have low mechanical properties, resistance to withdrawal of screws, hardness, and dimensional stability in response to changes in humidity.

In order to reduce the use of wood and replace it with other lignocellulosic materials in GPB, several investigations have been focused on using agricultural residues: banana fiber and rice straw [4], paper cellulose and abacá fiber [5,6], bagasse [7], jute and cabuya [8], sisal [9], recycled cellulose [10], and bagasse with natural rubber [11]. However, in general, these panels have low mechanical behavior and need to be reinforced with cement [12,13,14,15,16].

Currently, gypsum boards have a high market share since they are commonly used in the finishing of interior walls and ceilings. GPB made from agricultural residues is more environmentally friendly and could replace traditional gypsum boards. It is, however, necessary to improve their quality by making them more resistant and durable.

Giant reed (*Arundo donax* L.) is one of the largest types of grass growing in the Mediterranean region; it is a wild plant that grows annually, reaching average heights of 4 m and a mean thickness of 4 cm. It was used in building construction in many Mediterranean countries; however, it is today in disuse and has been replaced by different types of wood. Currently, reeds grow unchecked and at a great rate. When reeds colonize riverbanks, the riverside vegetation is impoverished, and infrastructures that cross watercourses are blocked, making drainage difficult. When the water level rises more than usual, this leads to flooding that causes environmental, economic, and material damage. These problems force the competent authorities to make significant economic investments in cleaning and clearing this plant waste [17]. Some investigations focused on valorizing giant reed with different adhesives such as cement [18], UF [19], citric acid [20], and starch [21].

Starch is a macroconstituent of many foods, being one of the most important plant products [22]. With a production of approximately 60 million tons, 60% is used in food and 40% is used in nonfood industries [23] as an additive in cement, paper, gypsum, adhesives, bioplastics, composites, etc. [24]. Due to their nature, starches have great potential as substitutes for synthetic polymeric materials for environmental purposes. In particleboards, they are used as a substitute for binders such as UF, phenol–formaldehyde (PF), and other petroleum derivatives.

The aim of the present research was to study the manufacturability of particleboards made from giant reed (*Arundo donax* L.) with gypsum plaster and starch following a method based on the wood industry dry process but with variations (no pretreatments, different parameters at the hot press, and adhesives) so that it can be produced in the particleboard industry and, therefore, counteract the high dependence on wood imports using an easily renewable resource such as giant reed. By controlling this reed, a more sustainable and formaldehyde-free particleboard can be obtained, which would reduce pressure on forest resources and create new job opportunities.

## 2. Materials and Methods

### 2.1. Materials

The materials used in the manufacturing of the particleboards were giant reed (*Arundo donax* L.), calcium sulfate hemihydrate (gypsum plaster), and different amounts of potato starch (*Solanum Tuberosum* L.).

Giant reed biomass was provided by the authorities in charge of clearing the banks of the Segura river, in southeast Spain. Reeds were cleaned from impurities and left outside for 12 months for air-drying (Figure 1). They were then cut and shredded in a blade mill. In order to obtain the particles, a vibrating sieve was used, and only those which passed through the 0.25 mm sieve were selected. Reed particles had a moisture content approximately of 8 wt.% and an apparent density of 537 ± 38 kg/m^3^. Particles were kept in ambient laboratory conditions with an average temperature of 20 °C and an approximate ambient humidity of 65%.

Commercial gypsum plaster (CaSO_4_ · ½H_2_O) without any additives with an apparent density of 2330 ± 20 kg/m³ was mixed with the giant reed. Potato starch from the food industry with a purity of 90% and a bulk density of 320 ± 15 kg/m³ was added to the gypsum and giant reed. This starch is a mixture of amylose and amylopectin (both polysaccharides), and it typically contains large oval granules and gels at a temperature of 58–65 °C.

### 2.2. Methods

#### 2.2.1. Board Manufacture

The method followed was based on the wood industry dry process but with variations (no pretreatments and different parameters at the hot press).

The experimental boards were manufactured with a 100:20 proportion by weight of giant reed/gypsum, adding different amounts of starch (0, 5, and 10 wt.%). Figure 2 shows the raw materials of a panel type 20:10.

First, particles of giant reed, starch, and gypsum were dry-mixed. Then, 10% water was sprayed in relation to the weight of the giant reed particles in a laboratory glue blender mixer with rotating blades and a volumetric capacity of 100 dm^3^ (model LGB100, IMAL S.R.L, Modena, Italy).

Afterward, the mat was formed manually in a mold with dimensions of 400 × 600 mm (Figure 3). Then, it was hot-pressed with the following processing parameters: 110 °C pressing temperature, 2.6 MPa pressing pressure, and three different pressing times (1, 2, and 3 h).

Particleboards were single-layered with an approximately thickness of 7 mm. Characteristics of the panels are shown in Table 1.

For each panel type and time, four replicates were made. Therefore, 36 particleboards were manufactured in total.

#### 2.2.2. Experimental Tests

The method followed was experimental. The tests were conducted in the Materials Laboratory of the Higher Technical College of Orihuela at the Miguel Hernández University of Elche.

In order to characterize the mechanical, physical, and thermal properties of each of the boards being studied, samples (Figure 4) were cut to the appropriate dimensions to carry out the tests [25] and to determine their values according to the European standards established for wood particleboards [26]. Tests performed were as follows: density [27], thickness swelling (TS) and water absorption (WA) after 2 and 24 h immersed in water [28], internal bonding strength (IB) [29], modulus of elasticity (MOE) and modulus of rupture (MOR) [30], and thermal conductivity [31].

The moisture content of the material was measured with a laboratory moisture meter (model UM2000, Imal S.R.L, Modena, Italy), the water immersion test was carried out in a heated tank, and the mechanical tests and density were performed with a universal testing machine (model IB700, Imal, S.R.L., Modena, Italy).

For the statistical analysis, SPSS v. 28.0 software (IBM, Chicago, IL, USA) was used. Analysis of variance (ANOVA) was performed, and, for the mean values of the tests, the standard deviation was obtained.

## 3. Results and Discussion

### 3.1. Physical Properties

Table 2 shows the mean results and the standard deviation of thickness, density, TS, and WA after 2 and 24 h, as well as the thermal conductivity of each type of particleboard manufactured.

Mean density of the boards ranged between 925 and 1095 kg/m^3^; therefore, they can be considered as high-density particleboards. The panels with the highest density were the 20:10 type, which included 10% starch in their manufacture. Longer times in the hot press resulted in more density.

Meanwhile, the 20:10 type had lower values of TS and WA than the remaining experimental panels. Increasing time in the hot press also improved the TS and WA results. This was probably related to the increase in density, since there were fewer gaps between the particles that could be filled with water.

Thermal conductivity seemingly decreased with the addition of starch and longer pressing times, but this could not be concluded. Panels manufactured with more than 5% starch for 2 h had the lowest conductivity values. Given the standard deviation of the results, it is possible that pressing the panels for more than 2 h would not have improved the results when starch was added.

### 3.2. Mechanical Properties

MOR values of the experimental panels can be seen in Figure 5. Increasing time in the hot press improved the results in types 20:0 and 20:5. In type 20:10, the standard deviation overlapped between 2 and 3 h of pressing time; therefore, this statement cannot be confirmed. The addition of starch resulted in better MOR results in all types, reaching a maximum in type 20:10 3 h with a mean value of 17.5 N/mm^2^.

The MOE results showed in Figure 6 followed a similar trend to MOR values but more pronounced. Increasing the pressing time and the starch proportion resulted in higher MOE values. Type 20:10 3 h reached 3196 N/mm^2^ of MOE.

From the results of IB in Figure 7, it can be concluded that increasing the proportion of starch resulted in better mechanical behavior in the three categories: MOR, MOE, and IB. Type 20:10 3 h had the highest IB value with 0.65 N/mm^2^. The standard deviation in types 20:0 and 20:5 of the results prevent concluding that increasing the pressing time between 2 and 3 h improved the results.

### 3.3. Statistical Analysis

Statistical analysis of Table 3 indicates that all the properties had dependency relationships (*p*-value < 0.05) with the panel type and the pressing time except for IB, which was only influenced by the panel type.

### 3.4. Discussion of the Results

The European standards [32] establish a minimum requirement for particleboards according to their thickness. Panels from 6 to 13 mm with an MOR value of 10.5 N/mm^2^ and an IB value of 0.28 N/mm^2^ can be considered as particleboards for general use in a dry environment (grade P1). Particleboards that can be used in the manufacture of furniture (grade P2) have to reach an MOR of 11.0 N/mm^2^, an MOE of 1800 N/mm^2^, and an IB of 0.40 N/mm^2^. For nonstructural boards in a humid environment, the minimum requirements are 15.0 N/mm^2^ for MOR, 2050 N/mm^2^ for MOE, 0.45 N/mm^2^ for IB, and 17% for TS at 24 h.

In the present study, boards manufactured with less than 10% starch did not reach the minimum mechanical values to be commercialized. Table 4 shows a comparison of type 20:10 results regarding European standards. Increasing the pressing time improved the performance of the particleboards. Type 20:10 with 1 h of pressing time could be classified as P1, whereas board types 20:10 with 2 and 3 h of pressing time reached the P2 grade. None of them could be classified as P3 since they did not meet the required TS value at 24 h. Further research is needed to take into account the density of the panels, the peak time, the amount of added starch, and the addition of hydrophobic substances in the manufacturing process.

The selection of a particle size of less than 0.25 mm for the manufacturing of the boards was based on our previous experience and the existing literature, since smaller particle sizes resulted in better properties [33].

The process followed in the research was slightly different since GPBs are not usually pressed or heated. Particleboards that follow a dry process usually require a pretreatment that consists of drying the material until it reaches a maximum of 4% moisture content. In the present study, there was no need for pretreatments, and the material could be used with a 8% moisture content.

Kim et al. [34] manufactured GPBs with *Eucalyptus* sp. and *Pinus massoniata* with citric acid (CA) with a targeted density of 1200 kg/m^3^ and dimensions of 300 × 300 × 10 mm. They reduced the moisture content of the GPB mats to about 2–3% in a dryer at 45 °C before pressing at room temperature the particleboards. Kim [35] used the same process for manufacturing particleboards with rice husk particles of 2 mm in different ratios (0%, 10%, 20%, 30%, and 40%). MOR and MOE values of the particleboards improved with the addition of rice husk with a peak content of 30% and then declined. IB reached its maximum with 20% and had a lower value with 40%. These two studies obtained similar results and are in line with commercial GPB, as shown in Table 5. The experimental boards of the present study achieved better mechanical results with the difference of applying heat in the pressing (110 °C) and only adding 20% gypsum.

This lower gypsum content is not enough for the binding of the particles. If the method used was cold pressing, the particles would have disaggregated. Therefore, since type 20:0 resulted in stable particleboards, some self-binding could have happened in the hot press within the giant reed in the absence of starch. The elucidation of this process is very complex with multiple theories such as the glass transition of cellulose, hemicellulose, and lignin [36], the starches, gluten, sorbitol, and sugars contained in the lignocellulosic material [37], and the presence of furfural [38,39].

In comparison with binderless particleboards, some authors indicated that a high temperature of at least 180 °C and high pressures of 3.5 MPa are needed in order to manufacture the boards [37]. Others concluded that, by extending the time in the hot press, better results can be achieved [40]. In the present investigation, the temperatures of the press (110 °C) and the pressure (2.1 MPa) were lower; therefore, this method consumes less energy and is more environment friendly. Good MOR and MOE values were obtained with a pressing time of 2 and 3 h. It may be possible for the mechanical behavior to be improved with longer pressing times; however, the energy required in the manufacture of the particleboards would increase, which is counterproductive, and the panels already reached the P2 grade values within 2 h.

## 4. Conclusions

It is feasible to manufacture particleboards made from giant reed, gypsum plaster, and starch that have good mechanical properties according to the European standards.

Through the addition of starch and increasing the pressing time of the experimental panels, better mechanical and physical properties were obtained. With 10% starch and a pressing time of 2 or 3 h, the panels reached the P2 classification for furniture production.

Using giant reed waste in manufacturing particleboards would be beneficial in terms of the environment and energy efficiency since less energy is required than the conventional process.

## Figures and Tables

**Figure 1 molecules-27-07305-f001:**
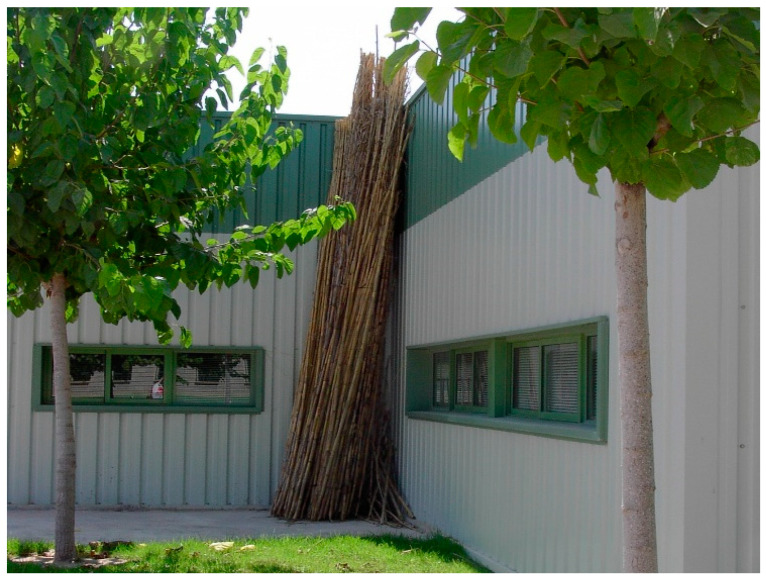
Giant reed left outside for 12 months for air drying.

**Figure 2 molecules-27-07305-f002:**
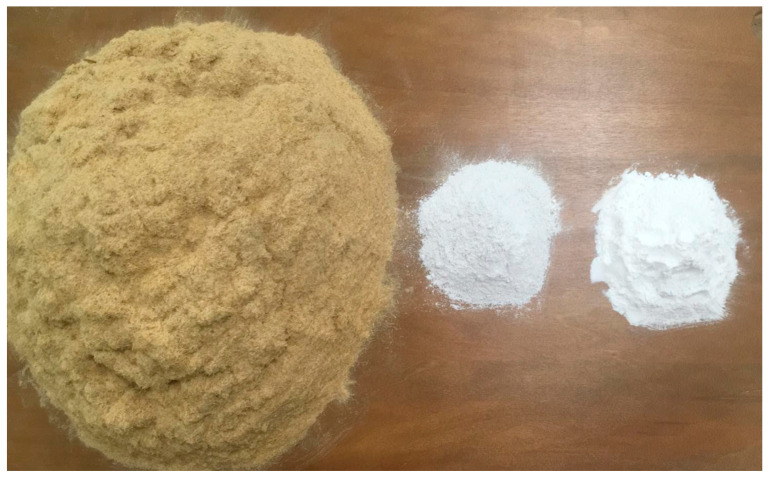
Giant reed (left), gypsum (center), and starch (right) particles used for manufacturing the experimental board type 20:10.

**Figure 3 molecules-27-07305-f003:**
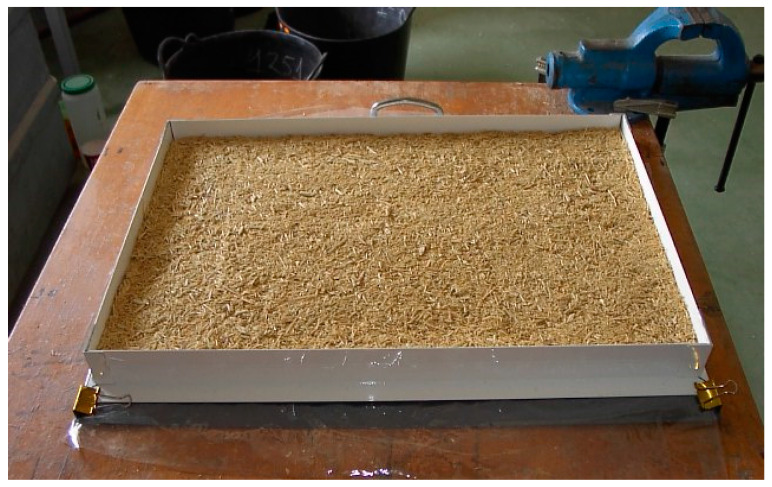
Mat formed manually to manufacture the particleboards.

**Figure 4 molecules-27-07305-f004:**
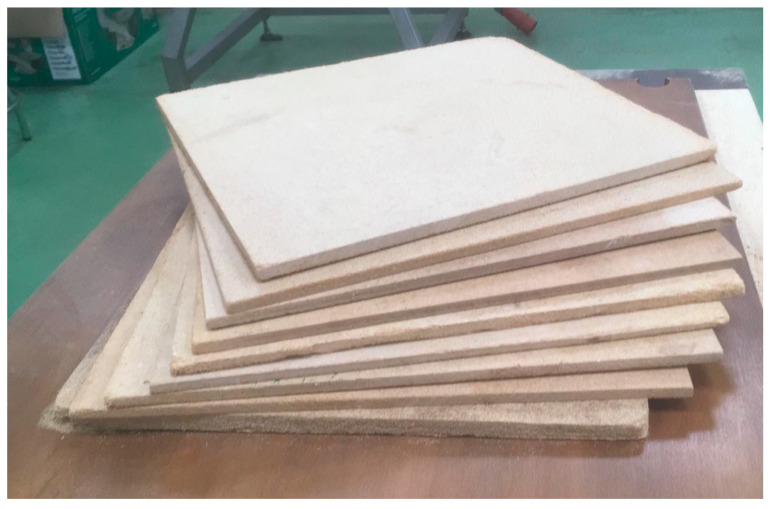
Samples of the experimental particleboards.

**Figure 5 molecules-27-07305-f005:**
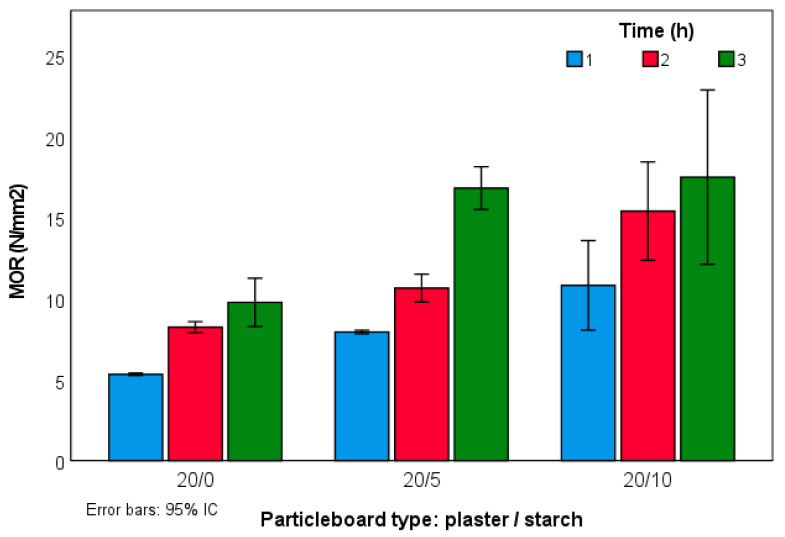
MOR values of the experimental panels.

**Figure 6 molecules-27-07305-f006:**
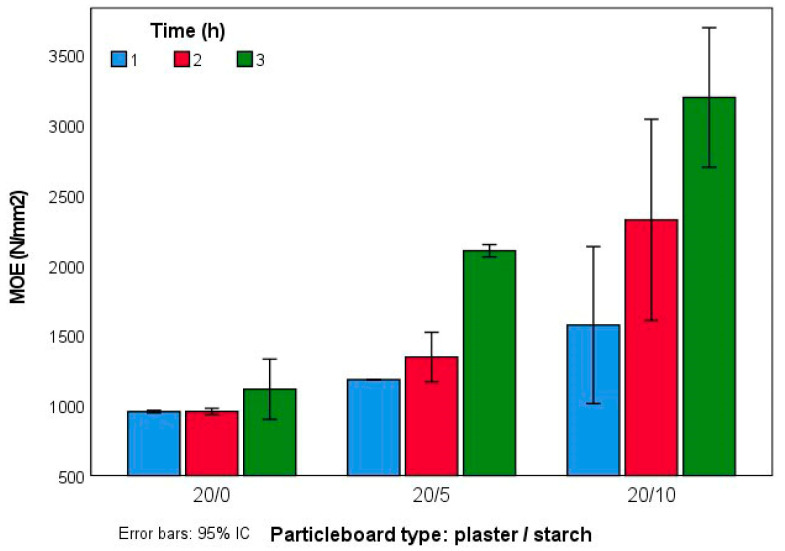
MOE values of the experimental panels.

**Figure 7 molecules-27-07305-f007:**
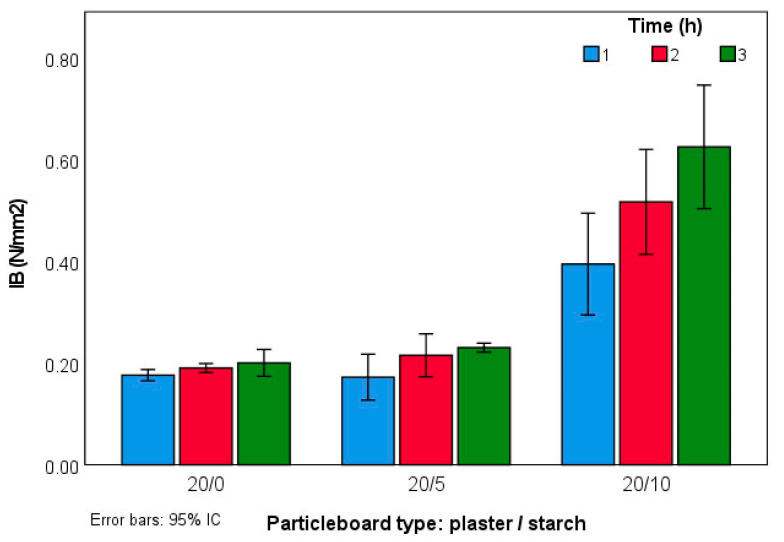
IB values of the experimental panels.

**Table 1 molecules-27-07305-t001:** Characteristics of the experimental panels.

Panel Type	Ratio Giant Reed/Gypsum Plaster/H_2_O in Weight	Ratio Giant Reed/Starch in Weight	Particle Size (mm)	Time (h)	Temperature (°C)
20:0	100:20:10	100:0	<0.25	1, 2, and 3	110
20:5		100:5			
20:10		100:10			

**Table 2 molecules-27-07305-t002:** Physical properties of the experimental panels.

Panel Type	Time (h)	Thickness (mm)	Density (kg/m^3^)	TS2h (%)	TS24h (%)	WA2h (%)	WA24h (%)	T. Cond. (W/mK)
20:0	1	7.70 (0.57)	925 (18)	29.16 (0.63)	47.60 (0.73)	75.75 (0.19)	84.19 (4.88)	0.072 (0.003)
	2	7.72 (0.34)	984 (28)	30.64 (1.91)	44.05 (3.87)	66.79 (2.84)	76.71 (0.12)	0.068 (0.005)
	3	7.11 (0.16)	994 (18)	31.32 (1.45)	39.66 (2.83)	63.66 (8.53)	75.11 (2.41)	0.064 (0.003)
20:5	1	7.36 (0.11)	929 (12)	37.06 (4.81)	44.47 (0.89)	74.14 (2.88)	85.53 (1.01)	0.063 (0.002)
	2	7.18 (0.14)	974. (59)	29.40 (3.12)	40.99 (1.45)	57.50 (7.58)	71.21 (4.23)	0.061 (0.002)
	3	6.89 (0.35)	1052 (11)	29.34 (0.00)	35.72 (3.93)	66.55 (0.35)	71.07 (5.28)	0.061 (0.001)
20:10	1	7.50 (0.55)	1024 (42)	25.72 (5.91)	36.16 (6.84)	48.96 (3.77)	58.85 (6.83)	0.066 (0.003)
	2	7.27 (0.41)	1053 (60)	26.34 (8.17)	32.19 (8.04)	43.84 (9.07)	57.67 (9.31)	0.063 (0.002)
	3	6.89 (0.14)	1095 (46)	23.19 (6.17)	28.37 (6.09)	41.47 (8.93)	52.25 (7.23)	0.061 (0.002)

Values in parentheses are the standard deviation.

**Table 3 molecules-27-07305-t003:** ANOVA of the results of the tests.

Factor	Properties	Sum of Squares	d.f.	Half Quadratic	F	*p*-Value
Panel Type	Density (kg/m^3^)	84,534.65	2	33,660.139	12.248	0.000
	TS 24 h (%)	1551.53	2	524.107	17.039	0.000
	WA 24 h (%)	1213.56	2	2254.850	44.949	0.000
	T. Cond. (W/mK)	0.001	2	0.001	11.57	0.000
	MOR (N/mm^2^)	15.08	2	175.442	14.294	0.000
	MOE (N/mm^2^)	754,422.53	2	6,978,605.131	23.840	0.000
	IB (N/mm^2^)	0.03	2	0.491	82.366	0.000
Pressing Time	Density (kg/m^3^)	115,661.14	2	29,069.446	9.781	0.000
	TS 24 h (%)	7.93	2	249.800	5.647	0.007
	WA 24 h (%)	38.76	2	417.170	2.984	0.042
	T. Cond. (W/mK)	0.001	2	0.001	5.580	0.007
	MOR (N/mm^2^)	13.36	2	166.638	13.118	0.000
	MOE (N/mm^2^)	130,082.85	2	3,131,162.189	6.518	0.000
	IB (N/mm^2^)	0.083	2	0.042	1.490	0.237

d.f.: degrees of freedom. F: Fisher–Snedecor distribution.

**Table 4 molecules-27-07305-t004:** Comparison of the properties of the experimental panels with 10% starch and the European standards [32].

Panel Type	MOR (N/mm^2^)	MOE (N/mm^2^)	IB (N/mm^2^)	WA 24 h (%)
20:10—1 h	10.82	1573	0.39	58.85
20:10—2 h	15.40	2324	0.52	57.67
20:10—3 h	17.49	3196	0.62	52.25
Grade P1 [32]	10.50	-	0.28	-
Grade P2 [32]	11.00	1800	0.40	-
Grade P3 [32]	15.00	2050	0.45	17.00

**Table 5 molecules-27-07305-t005:** Comparison with other particleboards made from gypsum and other lignocellulosic materials.

Material	Ratio Material/ Gypsum/H_2_O in Weight	Other Additions	Press. (MPa)	Time (h)	MOR (N/mm^2^)	MOE (N/mm^2^)	IB (N/mm^2^)	WA 24 h (%)
This study (type 20:10—2 h)	100:20:10	10% Starch	2.6	2	10.82	1573	0.39	58.85
Eucalyptus sp. [34]	30:100:40	0.05% CA	3	2	6.80	2700	0.24	29.50
Pinus massoniata [34]	30:100:40	0.05% CA	3	2	5.50	2100	0.43	28.00
Rice husk [35]	40:100:40	0.05% CA	3	2	6.70-	3800	0.34	17.00
Commercial GPB [34]	-	-	-	-	9.20	4500	0.30	28.00

## Data Availability

The data presented in this study are available within the article.

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
