# Peer review of "Assessment of the Properties of Giant Reed Particleboards Agglomerated with Gypsum Plaster and Starch"

_molecules, 2022, doi:10.3390/molecules27217305_

Round 1

Reviewer 1 Report

See attachment

Author Response

Dear reviewer,

Thank you very much for your feedback and your comments to improve our manuscript. We take note about avoiding the word “feasibility” since we do not want to confuse the reader and change it in the manuscript to “manufacturability” as you suggested.

We do not have the other comments though. Therefore, we kindly ask you if you could send them to us through the platform.

Best regards

Reviewer 2 Report

Authors need to consider the following comments to improve the submitted manuscript.

1. Introduction  section. Please reorganize the structure, and make paragraphs more concentrated on the topic, and sentences more concise.

2. Figure 1 and Figure 2 are just photos. More detailed information have to be included and compared, e.g., which treatment was used, images of materials before and after the treatment/processing, etc. 

3. While properties of the prepared environmentally friendly particle boards are characterized and discussed, comparison with other these green materials of similar kind has to be made, pointing out the advance of this study.

4. is the current work undertaking at experintal batch or larger pilot scale?

Round 2

Reviewer 2 Report

Authors have improved the manuscript to some extent, but I still have following concerns:

1. The objective now is changed to the manufacturability of particleboards made from giant reed (Arundo donax L.) with gypsum plaster and starch.  Literature short review describing current progress and shortcomings in this field has to be provided. Line89-90 one conclusive sentence is far from sufficient, especially when readers need to know the importance of this study just submitted.

2. The presentation of results has to be improved. For instance, consider to combine figures rather than individual ones, a combinatino of Fig.1+Fig. 2, Fig.3+Fig. 4.

3. It's necessary to give full name on specific words of their first appearance instead of abbreviation, e.g. MOE, MOR, IB in Abstract.
